# Stochastic Fractional Gradient Descent with Caputo $L_1$ Scheme for Deep Neural Networks

## Abstract

Stochastic gradient descent (SGD) has been used as a standard method to optimize deep neural networks (DNNs), where it essentially deals with first-order derivatives. Incorporating fractional derivatives into learning algorithms is expected to improve model performance, especially when the corresponding optimization problems involve objective functions with memory effects or long-range dependencies. The Caputo derivative is a fractional derivative that maintains consistency with integer-order calculus and produces more reliable solutions than other fractional derivatives, especially for differential equations. In this paper, we propose a novel Caputo-based SGD algorithm tailored for training DNNs. Our method exploits the Caputo $L_1$ scheme to achieve highly effective training and accurate prediction for large data by using gradient information from its past history to guide parameter updates in a more informed direction. This allows it to avoid local minima and saddle points, resulting in faster convergence to the target value. We conducted experiments on several benchmark datasets to evaluate our method. The results show that our method can improve the empirical performance over some traditional optimization methods in both accuracy and convergence.

## 1 Introduction

The performance of deep neural networks (DNNs) depends heavily on the choice of optimization algorithms. Stochastic gradient descent (SGD) (Bottou, 2010) has been used as a standard method for this purpose, and is known for its computational efficiency especially when handling large data. Several improved variants of SGD, such as Adam (Kingma & Ba, 2014), Yogi (Zaheer et al., 2018) and DiffGrad (Dubey et al., 2019) have been proposed and been employed in various situations of learning DNNs. Note that these algorithms essentially deal only with first-order derivatives.

On the other hand, fractional calculus, a branch of mathematical analysis that deals with non-integer order derivatives and integrals, which dates back to the 19th century (Lacroix, 1819; Leibniz, 1849), has gained recognition for its attractive properties such as long-term memory and nonlocality (Li & Zeng, 2015), and has been widely applied in diverse fields, including image processing, signal processing, and the analysis of neural networks (Kaslik & Sivasundaram, 2011; Rakkiyappan et al., 2016; Sun et al., 2018).

Some recent studies show that the integration of fractional calculus into optimization algorithms can lead to promising performance improvements in the training of DNNs (Wang et al., 2017; Sheng et al., 2020; Kan et al., 2021; Wang et al., 2022; Zhou et al., 2023; Shin et al., 2023; Altan et al., 2023; Wei et al., 2023). Prior to these, Pu et al. pioneered the incorporation of fractional calculus into gradient descent methods by directly replacing first-order derivatives in the methods with fractional-order derivatives (Pu et al., 2015). Thereafter, in Wang et al. (2017); Bao et al. (2018), the authors developed a fractional gradient descent (FGD) method for learning neural networks via back-propagation, where they use the Caputo derivative to calculate fractional-order gradients of loss functions. In Sheng et al. (2020), a fractional-order gradient approach for convolutional neural networks (CNNs) was proposed. Similarly in Taresh et al. (2022), the authors proposed a FGD method for CNNs to improve efficiency using a fixed memory step and an adjustable number of terms based on the Caputo definition. The authors of Yu et al. (2022) proposed a new fractional

order momentum (FracM) optimizer for DNNs using the Grünwald-Letnikov (GL) definition of adaptive optimization algorithms. They evaluated the FracM optimizer on CIFAR10/100 and IMDB datasets and achieved improved performance over the baseline optimizers. And, deep learning optimizers based on the GL derivative were proposed under the short-memory effect, named FCSGD_G_L and FCAdam_G_L, respectively (Zhou et al., 2023), which are fusions of SGD and Adam respectively with the GL derivative, allowing them to learn more effectively from past gradients.

In this paper, we propose a novel Caputo-based SGD algorithm tailored for training DNNs. Our method exploits the Caputo $L_1$ scheme in the calculation of fractional gradients. The Caputo $L_1$ scheme is an efficient way to calculate FGDs because it uses a linear combination of discrete differences to approximate the Caputo fractional derivative. This approximation scheme does not lose information about the local behavior of functions around the point where the derivative is calculated, resulting in a more accurate approximation than the existing Caputo-based FGD method. These properties of our method are expected to enhance the advantages of FGD methods over traditional gradient descent methods, such as robustness to noise and improved ability to learn long-range dependencies in data. Finally, we performed experiments on CIFAR-10/100 data using ResNet and VGG architectures to demonstrate the improved performance of our Caputo-based SGD method over existing ones.

The remainder of this paper is organized as follows. First, in Section 2, we give a brief review of fractional calculus and describe some related works. In Section 3, we provide the numerical approximation of the Caputo derivative with the Caputo $L_1$ scheme. In Section 4, we propose a stochastic FGD algorithm using the developed approximate Caputo derivative for learning DNNs. Finally, we show the numerical results of experiments on benchmark datasets in Section 5, and then conclude the paper in Section 6.

## 2 Background and Related Works

In this section, we give a brief overview of fractional calculus, and then highlight some of the latest advances in the field of FGD.

### 2.1 Fractional Calculus

The $q$-th derivative, which we denote by $D^q f$, is clearly defined when $q$ is a positive integer. Fractional calculus is a branch of mathematical analysis that deals with non-integer order derivatives and integrals. It is known that, while integer-order derivatives remain information about the rate of change of a function at a single point, fractional derivatives provide the ability to describe the behavior of a function that reflects the history of its trajectories (see, for example, Spanier & Oldham (1974)). This property is useful for describing the behavior of a function with complex memory effects or long-range interactions.

Although the mathematical debate over the definition of the fractional derivative has not been settled, the following three definitions are widely accepted by researchers: The Grünwald–Letnikov (GL), the Riemann-Liouville (RL), and the Caputo derivatives (Kilbas et al., 2006). Despite the differences in these definitions, they are all widely used in research and have their unique advantages and limitations.

**Definition 1** (GL derivative). The Grünwald–Letnikov (GL) fractional derivative with fractional-order $\alpha > 0$ of the given function $f(w)$ is defined as

$$_{GL}D_{a,w}^{\alpha}f(w) = \lim_{h \to 0} \frac{1}{h^{\alpha}} \sum_{k=0}^{L} (-1)^k \begin{pmatrix} \alpha \\ k \end{pmatrix} f(w - kh),$$

where, $L = [\frac{(w-a)}{h}]$, and $\begin{pmatrix} p \\ q \end{pmatrix} = \frac{\Gamma(p+1)}{\Gamma(q+1)\Gamma(p-k+1)}$, $p \in \mathbb{R}$, $q \in \mathbb{N}$ denotes the binomial coefficient. $\Gamma(\cdot)$ is the Euler's gamma function, $\Gamma(\tau) = \int_0^{\infty} w^{\tau-1}e^{-w}dw$.

**Definition 2** (RL derivative). The Riemann-Liouville (RL) fractional derivative with fractional-order $\alpha > 0$ of the given function $f(w)$ is defined as

$$_{RL}D_{a,w}^{\alpha}f(w) = \frac{1}{\Gamma(n-\alpha)} \frac{d^n}{dt^n} \int_a^w (w-\tau)^{n-\alpha-1}f(\tau)d\tau,$$

where $n$ is a positive integer satisfying $n - 1 \leq \alpha < n$.

**Definition 3** (Caputo derivative). The Caputo fractional derivative with fractional-order $\alpha > 0$ of the given function $f(w)$ is defined as

$$_C D^\alpha_{a,w} f(w) = \frac{1}{\Gamma(n-\alpha)} \int_a^w (w-\tau)^{n-\alpha-1} f^{(n)}(\tau) d\tau, \tag{1}$$

where $n$ is a positive integer satisfying $n - 1 < \alpha \leq n$. The corresponding discreate expansion of Eq. (1) is as follows (Sheng et al., 2020):

$$_C D^\alpha_{a,w} f(w) = \sum_{k=n}^\infty \frac{f^{(k)}(w)}{\Gamma(k+1-\alpha)} (w-a)^{k-\alpha}.$$

For $\alpha \in (0, 1)$, we have

$$_C D^\alpha_{a,w} f(w) = \frac{1}{\Gamma(1-\alpha)} \int_a^w (w-\tau)^{-\alpha} f^{(1)}(\tau) d\tau. \tag{2}$$

And the following gives the chain rule for the Caputo derivative:

**Definition 4.** (Wang et al., 2017) Suppose $f(g(x))$ to be a composite function, then the $\alpha$-th order Caputo derivative with respect to $x$ is

$$_C D^\alpha_{a,x} f(g(x)) = \frac{\partial f(g)}{\partial g} \cdot {_C D^\alpha_{a,x}} g(x). \tag{3}$$

### 2.2 Fractional Gradient Descent

As is mentioned above, fractional derivatives provide the ability to describe the behavior of a function that reflects the history of its trajectories. Thus, their incorporation into gradient descent methods is expected to guide parameter updates in a more informed direction. Basically, such an FGD method can be achieved by replacing a conventional gradient in the update law with a fractional one, i.e., $w_{k+1} = w_k - \vartheta D^\alpha_{a,w_k} f(w_k)$, where $\vartheta$ is the learning rate, $D^\alpha_{a,w_k}$ is the fractional derivative operator and $\alpha > 0$ is the fractional-order of the derivative. $D^\alpha_{a,w_k}$ is determined based on the definitions of fractional derivatives such as the GL, RL and Caputo derivatives (see Definitions 1, 2, and 3) and their numerical schemes. Recently, there has been an increasing interest in the analysis of FGD approaches (Pu et al., 2015; Bao et al., 2018; Wei et al., 2019; Lou et al., 2022; Taresh et al., 2022; Zhou et al., 2023).

The authors in Pu et al. (2015) have proposed the FGD optimization approach for NNs. The update law was designed as follows:

$$w_{k+1} = w_k - \vartheta {_C D^\alpha_{a,w_k}} f(w_k), \tag{4}$$

where $_C D^\alpha_{a,w_k}$ is obtained based on the following discrete form of the Caputo derivative:

$$_C D^\alpha_{a,w} f(w) = \sum_{k=n}^\infty \binom{\alpha-n}{k-n} \frac{f^{(k)}(w)}{\Gamma(k+1-\alpha)} (w-a)^{k-\alpha}. \tag{5}$$

Bao et al. have also proposed the FGD with the Caputo derivative for DNNs (Bao et al., 2018), where the property of the fractional differentiation of power function $_C D^\alpha_{a,w_k}(w-a)^s = \frac{\Gamma(s+1)}{\Gamma(s-\alpha+1)}(w-a)^{s-\alpha}$, $s > -1$ was utilized to approximate Caputo derivatives.

The value of the lower terminal $a$ may affect the convergent results, which were different from the real extreme points when using the approach (4). Therefore, the authors in Wei et al. (2020) rewrote the update law (4) by truncating higher order terms of Eq. (5) for $0 < \alpha < 1$ as

$$_C D^\alpha_{a,w_k} f(w_k) = \frac{f^{(1)}(w_k)}{\Gamma(2-\alpha)} |(w_k - a) + \varepsilon|^{1-\alpha},$$

where $\varepsilon$ represents the small non-negative number to avoid the non-convergence when $w_k = a$. Meanwhile, in Taresh et al. (2022), the authors proposed to iterate the value of the lower terminal of (4) to obtain the real extreme point of $w$ with the modified update law:

$$w_{k+1} = w_k - \vartheta \sum_{k=1}^{M} \frac{f^{(1)}(w_{k-1})}{\Gamma(n+1-\alpha)}|(w_k - w_{k-1}) + \varepsilon|^{n-\alpha},$$

where $n < M$. And more recently, the fractional SGD and fractional Adam algorithms have been proposed in Zhou et al. (2023) by replacing the fractional derivative in Eq. (4) with the one of Definition 1.

## 3 Numerical Approximation for Caputo Derivative

In this paper, we propose a novel FGD algorithm based on the Caputo derivative. The Caputo derivative is a recognized technique for tackling physical phenomena, especially initial value problems because it adheres to the fundamental principles of integer-order calculus, where the fractional derivatives of a constant function equals zero (Kilbas et al., 2006). Unlike prior methods, we employ the Caputo $L_1$ numerical scheme to compute the fractional gradient update rule.

### 3.1 Caputo Derivative with $L_1$ scheme

The Caputo derivative (2) can be discretized based on the $L_1$ approximation, which relies on a linear interpolation formula for functions within each subinterval. Let us consider the fractional differential equations as follows:

$$_C\mathcal{D}_{0,w}^{\alpha} Y(w) = f(Y(w)), \quad 0 < \alpha < 1. \tag{6}$$

Assume that $Y(w)$ is the solution of (6) on the interval $w \in \Xi := [0, T]$, where $T > 0$. Discretizing the domain $\Xi$ uniformly with step size $\Delta w = w_{i+1} - w_i$ as follows: $\Xi_N := \{w_i : 0 = w_0 < \cdots < w_1 < \cdots < w_i < w_{i+1} < \cdots < w_n = T\}$, where $w_i = i\Delta w$.

For $i = 0, ..., n-1$, the discretization of (6) is given by using the $L_1$-scheme:

$$
\begin{aligned}
_C D_{0,w}^{\alpha} Y(w)|_{w=w_n} &= \frac{1}{\Gamma(1-\alpha)} \int_{w_0}^{w_n} (w_n - \tau)^{-\alpha} Y^{(1)}(\tau) d\tau \\
&= \frac{1}{\Gamma(1-\alpha)} \sum_{i=0}^{n-1} \int_{w_i}^{w_{i+1}} (w_n - \tau)^{-\alpha} Y^{(1)}(\tau) d\tau \\
&\approx \frac{-1}{\Gamma(1-\alpha)} \sum_{i=0}^{n-1} \frac{Y(w_{i+1}) - Y(w_i)}{\Delta w} \left[ \frac{(w_n - w_{i+1})^{-\alpha+1} - (w_n - w_i)^{-\alpha+1}}{-\alpha + 1} \right] \\
&= \sum_{i=0}^{n-1} \delta_{n-i-1} \left( Y(w_{i+1}) - Y(w_i) \right),
\end{aligned}
\tag{7}
$$

where $\delta_k = \frac{(\Delta w)^{-\alpha}}{\Gamma(2-\alpha)}[(k+1)^{1-\alpha} - k^{1-\alpha}]$. This method necessitates the storage of all previous function values, creating a bottleneck for long-term simulations.

To visually represent the improved convergence of the suggested FGD (4) under the Caputo $L_1$ scheme (7), the quadratic function—a popular kind of objective function—is selected. Let us assume that, $\vartheta = 0.08$ (learning rate), $\mu = 0.4$ (momentum), $\alpha = 0.8$ (fractional-order), and $\Delta w = 0.01$ (step-size), we evaluate the performance of Caputo-based SGD with momentum against two other widely adopted approaches - gradient descent and SGD with momentum when applied to a basic quadratic objective function $f(x, y) = 10x^2 + y^2$. The starting point for the optimization process is at $(x, y) = (1, -10)$, with the origin as the function's minimizer. The comparison in Figure 1 clearly demonstrates that our proposed Caputo-based SGD exhibits a superior convergence rate in achieving the desired optimization objective, outperforming both the conventional gradient descent and SGD with momentum methodologies.

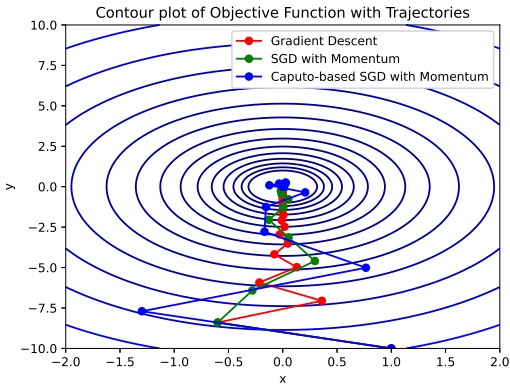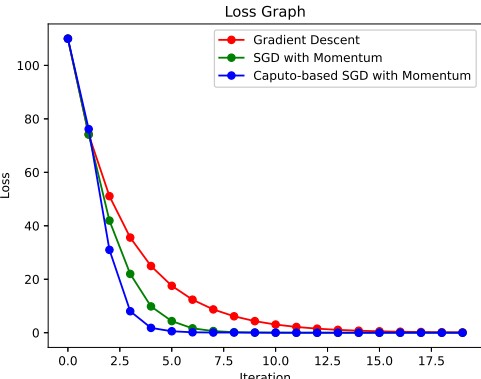

Figure 1: Convergence of descent approaches (Gradient Descent, SGD with Momentum, and Caputo-based SGD with Momentum) on functions $f(x, y) = 10x^2 + y^2$ starting at $(x, y) = (1, -10)$. The trajectory of the objective functions is shown in the contour graph (left) and loss graph (right).

## 4 DNNs with Caputo-based FGD optimization

In this section, we establish a simple structure of DNNs that features end-to-end connections for all layers. Following that, we propose the Caputo-based SGD with the help of the Caputo $L_1$ scheme. Next, we establish theoretical results by proving the convergence of the proposed method and its associated truncation error. Then optimizing the addressed DNNs through the back-propagation algorithm which updates the parameters of networks based on Caputo-based SGD.

### 4.1 Structure of DNNs

The simple structure of DNNs and how they work are discussed briefly as follows:

- Input layer: $H_0 = x_t$

- Hidden layer: $H_l = \varsigma_l(\mathcal{A}_l)$, $\mathcal{A}_l = w_l H_{l-1} + b_l$, $l = 1, 2, \cdots, L$.

- Output layer: $H(x_t, \rho) = w_{L+1} H_L + b_{L+1}$.

Here, $L$ denotes the number of hidden layers in NN. $n_l$, $l = 0, 1, 2, \cdots, L+1$, is the number of neurons in the layer $l$, the weights and bias of the hidden layer $l$ are depicted as $w_l \in \mathrm{R}^{n_l, n_{l-1}}$ and $b_l \in \mathrm{R}^{n_l}$. $\varsigma_l(\cdot)$ represents the activation functions of the $l$-th hidden layer, and $H_l$ denotes the output of the hidden layer $l$. $\rho$ is the unknown parameter vector to be calculated.

The process of training NNs consists of two fundamental steps, including forward propagation and backward propagation. In this paper, we aim to explore the use of FGD as a replacement for the traditional method of training NNs via backward propagation. We believe that this alternative approach could offer significant benefits, and we look forward to presenting our findings.

### 4.2 Caputo-based SGD approach

We aim to improve the accuracy, and speed of convergence, and minimize the total loss/error values of the DNNs. To this end, the weights and biases are updated by using the Caputo-based FGD. Then the updating formula at the iteration $k \in \mathbb{N}$ is defined as

$$
\begin{aligned}
(w_l)_{k+1} &= (w_l)_k - \vartheta_C D^\alpha_{(w_l)_k} \mathcal{E}, \\
(b_l)_{k+1} &= (b_l)_k - \vartheta_C D^\alpha_{(b_l)_k} \mathcal{E},
\end{aligned}
\tag{8}
$$

where $\vartheta$ denotes the learning rate, and $\mathcal{E}$ is the loss/objective function of DNNs.

**Proposition 1.** For any DNN with a loss/objective function $\mathcal{E}$, the fractional-order Caputo derivative of $\mathcal{E}$ with respect to the DNN parameters (weights $w_l$ and biases $b_l$) can be expressed as follows:

$$
\begin{aligned}
{}_C D^\alpha_{(w_l)}\mathcal{E} &= \frac{\partial \mathcal{E}}{\partial H_l}\varsigma'_l(\mathcal{A}_l)H_{l-1}\sum_{s=0}^{n-1}\delta_{n-s-1}\left(w_l^{s+1}-w_l^s\right), \\
{}_C D^\alpha_{(b_l)}\mathcal{E} &= \frac{\partial \mathcal{E}}{\partial H_l}\varsigma'_l(\mathcal{A}_l)\sum_{s=0}^{n-1}\delta_{n-s-1}\left(b_l^{s+1}-b_l^s\right),
\end{aligned}
\tag{9}
$$

where $\delta_r = \frac{(\Delta w)^{-\alpha}}{\Gamma(2-\alpha)}\left[(r+1)^{1-\alpha}-r^{1-\alpha}\right]$, and $\Delta w$ is the step size used in the Caputo $L_1$ approximation. To simplify the notation, we can replace $w_{l(s+1)}$, $w_{l(s)}$, $b_{l(s+1)}$, and $b_{l(s)}$ with $w_l^{s+1}$, $w_l^s$, $b_l^{s+1}$, and $b_l^s$, respectively.

*Proof.* According to (3), one can obtain the Caputo fractional derivative of $\mathcal{E}$ with respect to the DNNs parameters (weights and bias) as follows:

$$
\begin{aligned}
{}_C D^\alpha_{(w_l)}\mathcal{E} &= \frac{\partial \mathcal{E}}{\partial \mathcal{A}_l}{}_C D^\alpha_{(w_l)}\mathcal{A}_l, \\
{}_C D^\alpha_{(b_l)}\mathcal{E} &= \frac{\partial \mathcal{E}}{\partial \mathcal{A}_l}{}_C D^\alpha_{(b_l)}\mathcal{A}_l,
\end{aligned}
\tag{10}
$$

From the structure of DNNs, the integer-order gradient term $\frac{\partial \mathcal{E}}{\partial \mathcal{A}_l}$ is expressed as

$$
\begin{aligned}
\frac{\partial \mathcal{E}}{\partial \mathcal{A}_l} &= \frac{\partial \mathcal{E}}{\partial H_l}\frac{\partial H_l}{\partial \mathcal{A}_l} = \frac{\partial \mathcal{E}}{\partial H_l}\varsigma'_l(\mathcal{A}_l), \\
\frac{\partial \mathcal{E}}{\partial H_{l-1}} &= \frac{\partial \mathcal{E}}{\partial H_l}\frac{\partial H_l}{\partial \mathcal{A}_l}\frac{\partial \mathcal{A}_l}{\partial H_{l-1}} = \frac{\partial \mathcal{E}}{\partial \mathcal{A}_l}w_l.
\end{aligned}
\tag{11}
$$

By substituting (11) into (10), we have

$$
\begin{aligned}
{}_C D^\alpha_{(w_l)}\mathcal{E} &= \frac{\partial \mathcal{E}}{\partial H_l}\varsigma'_l(\mathcal{A}_l){}_C D^\alpha_{(w_l)}\mathcal{A}_l, \\
{}_C D^\alpha_{(b_l)}\mathcal{E} &= \frac{\partial \mathcal{E}}{\partial H_l}\varsigma'_l(\mathcal{A}_l){}_C D^\alpha_{(b_l)}\mathcal{A}_l.
\end{aligned}
\tag{12}
$$

Based on the Caputo $L_1$ scheme (7),

$$
\begin{aligned}
{}_C D^\alpha_{(w_l)}\mathcal{A}_l &= H_{l-1}\sum_{s=0}^{n-1}\delta_{n-s-1}\left(w_l^{s+1}-w_l^s\right), \\
{}_C D^\alpha_{(b_l)}\mathcal{A}_l &= \sum_{s=0}^{n-1}\delta_{n-s-1}\left(b_l^{s+1}-b_l^s\right),
\end{aligned}
\tag{13}
$$

where $\delta_r = \frac{(\Delta w)^{-\alpha}}{\Gamma(2-\alpha)}\left[(r+1)^{1-\alpha}-r^{1-\alpha}\right]$. From (12) and (13), the fractional-order updating gradient descent (10) can be rewritten as

$$
\begin{aligned}
{}_C D^\alpha_{(w_l)}\mathcal{E} &= \frac{\partial \mathcal{E}}{\partial H_l}\varsigma'_l(\mathcal{A}_l)H_{l-1}\sum_{s=0}^{n-1}\delta_{n-s-1}\left(w_l^{s+1}-w_l^s\right), \\
{}_C D^\alpha_{(b_l)}\mathcal{E} &= \frac{\partial \mathcal{E}}{\partial H_l}\varsigma'_l(\mathcal{A}_l)\sum_{s=0}^{n-1}\delta_{n-s-1}\left(b_l^{s+1}-b_l^s\right).
\end{aligned}
\tag{14}
$$

$\square$

The following theorems illustrate the convergence and the truncation error analysis of objective function $\mathcal{E}$ with Caputo $L_1$ scheme.

**Theorem 1.** Let $(\zeta^l)_k$ be the layers of DNNs updated by Caputo-based SGD (8) converge to the real extreme point $\zeta_l^*$ if and only if there exists $\epsilon > 0$ and a learning rate $\vartheta$ such that for all $k$, the following inequality holds:

$$\|e_{k+1}\| \leq (1 - \vartheta\epsilon)^k \|e_0\|,$$

where $e_k = (\zeta^l)_k - \zeta_l^*$, $\epsilon = \kappa \frac{(\Delta\zeta)^{-\alpha}}{\Gamma(2-\alpha)} \sum_{s=0}^{n-1} \left[ (n-s)^{1-\alpha} - (n-s-1)^{1-\alpha} \right]$.

*Proof.* Let $e_k = (\zeta^l)_k - \zeta_l^*$. Then the update law (8) can be expressed as:

$$e_{k+1} = e_k - \vartheta_C D_{0,(\zeta_l)_k}^\alpha \mathcal{E}((\zeta_l)_k)_n)$$

According to (9), one can get

$$e_{k+1} = e_k - \vartheta \frac{\partial\mathcal{E}}{\partial(\zeta_l)_{k-1}} \sum_{s=0}^{n-1} \delta_{n-s-1} \left( (\zeta_l)_k^{s+1} - (\zeta_l)_k^s \right), \tag{15}$$

where $\frac{\partial\mathcal{E}}{\partial(\zeta_l)_{k-1}} = \frac{\partial\mathcal{E}}{\partial H_l} \varsigma_l'(\mathcal{A}_l) H_{l-1}$. It follows that, there exists a sufficiently large number $\mathbf{N} \in \mathbb{N}$, such that for any $k-1 > N$, then $\varrho = \inf_{k-1 > \mathbf{N}} \left| \frac{\partial\mathcal{E}}{\partial(\zeta_l)_{k-1}} \right| > 0$ is guaranteed. From (15), the following results can be obtained:

$$\|e_{k+1}\| \leq \|e_k\| + \vartheta \left| \frac{\partial\mathcal{E}}{\partial(\zeta_l)_{k-1}} \right| \sum_{s=0}^{n-1} \delta_{n-s-1} \| \left( (\zeta_l)_k^{s+1} - (\zeta_l)_k^s \right) \|$$

$$\leq \|e_k\| + \vartheta\varrho \sqrt{\sum_{s=0}^{n-1} \delta_{n-s-1}^2 \sum_{s=0}^{n-1} \left( \|(\zeta_l)_k^{s+1} - (\zeta_l)_k^s\| \right)^2}$$

$$\leq \|e_k\| + \vartheta\varrho \sqrt{\sum_{s=0}^{n-1} \delta_{n-s-1}^2 \sum_{s=0}^{n-1} \|e_k\|^2}$$

$$\leq \|e_k\| + \vartheta\varrho \sqrt{\sum_{s=0}^{n-1} \delta_{n-s-1}^2 \sum_{s=0}^{n-1} \|e_k\|^2}$$

$$\leq \|e_k\| + \vartheta\varrho \sqrt{\left( \sum_{s=0}^{n-1} \delta_{n-s-1}^2 \right) \|e_k\|}$$

$$\leq \left( 1 + \vartheta\varrho \sqrt{\sum_{s=0}^{n-1} \delta_{n-s-1}^2} \right) \|e_k\|$$

$$\leq (1 + \vartheta\epsilon) \|e_k\|,$$

where $\epsilon = \varrho \sqrt{\sum_{s=0}^{n-1} \delta_{n-s-1}^2}$, and $\delta_r = \frac{(\Delta\zeta)^{-\alpha}}{\Gamma(2-\alpha)} \left[ (r+1)^{1-\alpha} - (r)^{1-\alpha} \right]$. Now, for all $k$,

$$\|e_{k+1}\| \leq (1 + \vartheta\epsilon)^k \|e_0\|.$$

Next, analyze the convergence rate by rewriting the above inequality with the help of a natural logarithm:

$$\ln(\|e_{k+1}\|) \leq k \ln(1 + \vartheta\epsilon) + \ln(\|e_0\|)$$

Based on Taylor's series expansion, $\ln(1 + \vartheta\epsilon) \approx \vartheta\epsilon$ This is a valid approximation for small $\vartheta\epsilon$. So, we have:

$$\ln(\|e_{k+1}\|) \leq k\vartheta\epsilon + \ln(\|e_0\|)$$

Taking the exponential on both sides, one can get

$$\|e_{k+1}\| \le e^{k\vartheta\epsilon}\|e_0\|$$

Thus, $(\zeta^l)_k$ converges to $\zeta_l^*$ exponentially with a rate of $e^{k\vartheta\epsilon}$. If $0 < \vartheta\epsilon < 1$, the error will decrease with each iteration, and the proposed method is convergent. The value of $\epsilon$ depends on the Caputo $L_1$ scheme with the convergence rate $(2-\alpha)$ or $O(\Delta\zeta^{2-\alpha})$. Achieving an optimal convergence rate requires careful adjustment of both the learning rate ($\vartheta$) and the parameter $\epsilon$. The $\vartheta\epsilon$ is closer to zero, the faster convergence. However, choosing a very small $\vartheta$ may slow down the convergence due to small step sizes. This completes the proof of the convergence of the proposed Caputo-based FGD with the Caputo $L_1$ scheme. $\qquad\square$

**Theorem 2.** For any $0 < \alpha < 1$, the $L_1$ approximation is defined as (7). Then, there exists a positive constant $\kappa$, $\kappa = \min\limits_{0 \le i \le n-1} |\partial\mathcal{E}/\partial\zeta_l^i|$. The truncation error $R_i = {}_C D^\alpha_{0,\zeta_l}\mathcal{E}(\zeta_l)|_{\zeta_l=\zeta_l^i} - {}_C\mathcal{D}^\alpha_{0,\zeta_l}\mathcal{E}(\zeta_l)|_{\zeta_l=\zeta_l^i}$, $i = 0, ..., n-1$, satisfies:

$$|R_i| \le \mathcal{C}(\Delta\zeta)^{2-\alpha} \max_{t \in [\zeta_l^i, \zeta_l^{i+1}]} |\mathcal{A}''(t_l)|, \tag{16}$$

where $\mathcal{C} = \frac{\alpha\kappa}{2\Gamma(3-\alpha)}$.

*Proof.* For $i = 1, 2, \cdots, n-1$, $\varpi_{1,i}\mathcal{A}(\zeta_l) = \mathcal{A}(\zeta_l^{i-1})\frac{\zeta_l^i-\zeta_l}{\Delta\zeta} + \mathcal{A}(\zeta_l^i)\frac{\zeta_l-\zeta_l^{i-1}}{\Delta\zeta}$. Then, following theory of piecewise linear interpolation, $\mathcal{A}(\zeta_l) - \varpi_{1,i}\mathcal{A}(\zeta_l) = \frac{\mathcal{A}''(s_l^i)}{2}(\zeta_l^i - \zeta_l)(\zeta_l - \zeta_l^{i-1})$, $s_l \in (\zeta_l^{i-1}, \zeta_l^i)$, one can obtain

$$
\begin{aligned}
R_i &= {}_C D^\alpha_{0,\zeta_l}\mathcal{E}(\zeta_l)|_{\zeta_l=\zeta_l^i} - {}_C\mathcal{D}^\alpha_{0,\zeta_l}\mathcal{E}(\zeta_l)|_{\zeta_l=\zeta_l^i}\\
&= \frac{1}{\Gamma(1-\alpha)}\frac{\partial\mathcal{E}}{\partial(\zeta_l)}\int_{\zeta_l^0}^{\zeta_l^i}(\zeta_l^i - \tau)^{-\alpha}\left(\varpi_{1,i}\mathcal{A}(\tau) - \mathcal{A}(\tau)\right)' d\tau\\
&= \frac{1}{\Gamma(1-\alpha)}\frac{\partial\mathcal{E}}{\partial(\zeta_l)}\left[(\varpi_{1,i}\mathcal{A}(\tau) - \mathcal{A}(\tau))(\zeta_l^i - \tau)^{-\alpha}|_{\zeta_l^0}^{\zeta_l^i} - \alpha\int_{\zeta_l^0}^{\zeta_l^i}(\zeta_l^i - \tau)^{-\alpha-1}(\varpi_{1,i}\mathcal{A}(\tau) - \mathcal{A}(\tau))d\tau\right]\\
&= \frac{1}{\Gamma(1-\alpha)}\frac{\partial\mathcal{E}}{\partial(\zeta_l)}\left[-\frac{\mathcal{A}''(s_l^i)}{2}(\tau - \zeta_l)(\zeta_l^i - \tau)^{1-\alpha}|_{\zeta_l^0}^{\zeta_l^i} + \alpha\int_{\zeta_l^0}^{\zeta_l^i}(\zeta_l^i - \tau)^{-\alpha}\frac{\mathcal{A}''(s_l^i)}{2}(\tau - \zeta_l^0)d\tau\right]\\
&= \left|\frac{\alpha\kappa}{2\Gamma(1-\alpha)}\mathcal{A}''(t_l)\int_{\zeta_l^0}^{\zeta_l^i}(\zeta_l^i - \tau)^{-\alpha}(\tau - \zeta_l^0)d\tau\right|\\
|R_i| &\le \mathcal{C}(\Delta\zeta)^{2-\alpha}\max_{t_l \in [\zeta_l^i, \zeta_l^{i+1}]}|\mathcal{A}''(t_l)|.
\end{aligned}
$$

This completes the proof of the truncation error (16). $\qquad\square$

## 5 Numerical Experiments

The goal of this section is to show the performance of the Caputo-based SGD optimization method as described in Algorithm 1, using the Caputo $L_1$ numerical approximation. For this purpose, we report the results of our experiments on image recognition accuracy with the CIFAR-10 and CIFAR-100 datasets, which are collections of images from 10 and 100 classes respectively. We use two deep learning architectures, ResNet and VGG, which are well-known for their high accuracy on image classification tasks. These experiments were conducted using GPUs, specifically the Tesla V100-SXM2-32GB. The deep learning models were developed in Python 3.10.12, utilizing PyTorch 2.0.1, on the Windows 11 Home 64-bit operating system.

---

**Algorithm 1** The Caputo-based SGD algorithm

---

**Input:** $\vartheta$ (initial learning rate), $\zeta_0$ (params), $\mathcal{E}(\zeta)$ (objective), $0 < \alpha < 1$ (fractional-order), *maximize=False*, *nesterov=False*, $\Delta\zeta$ (step-size), $\lambda = 0$ (weight decay), $\mu = 0$ (momentum), $\tau = 0$ (dampening)

**for** $k = 1$ **to** $\cdots$ **do**
    $\nabla_k \leftarrow {_C}D^\alpha_{(\zeta_k)}\mathcal{E}(\zeta_{k-1})$    (FGD based on (9))
    **if** $\lambda \neq 0$ **then**
        $\nabla_k \leftarrow \nabla_k + \lambda\zeta_{k-1}$
    **end if**
    **if** $\mu \neq 0$ **then**
        **if** $k \geq 1$ **then**
            $\mathcal{M}_k \leftarrow \mu b_{k-1} + (1 - \tau)\nabla_k$
        **else**
            $\mathcal{M}_k \leftarrow \nabla_k$
        **end if**
        **if** *nesterov* **then**
            $\nabla_k \leftarrow g_k + \mu\mathcal{M}_k$
        **else**
            $\nabla_k \leftarrow \mathcal{M}_k$
        **end if**
    **end if**
    $\vartheta_k \leftarrow \phi(k) \cdot \vartheta$
    **if** *maximize* **then**
        $\zeta_k \leftarrow \zeta_{k-1} + \vartheta_k\nabla_k$
    **else**
        $\zeta_k \leftarrow \zeta_{k-1} - \vartheta_k\nabla_k$
    **end if**
**end for**
**return** $\zeta_k$

---

## 5.1 Datasets and Models

The CIFAR-10 and CIFAR-100 datasets (Krizhevsky et al., 2009) are widely utilized in computer vision and image classification endeavors. These datasets consist of $32 \times 32$ color images that are evenly distributed, making them exceptionally well-suited for training and assessing image classification models. CIFAR-10 comprises 10 mutually exclusive categories, whereas CIFAR-100 comprises 100 categories that are sorted into 20 superclasses.

CNNs are the most popular deep learning model for image classification. They use a series of convolutional layers to extract features from images. Based on their architecture, CNNs can be classified into different types, including residual networks (ResNets) (He et al., 2016) and visual geometry groups (VGG) (Simonyan & Zisserman, 2014). ResNets and VGG are deep, efficient, and adaptable image classification models. In this paper, we train and validate our proposed optimization strategy on the CIFAR-10 and CIFAR-100 datasets using ResNet and VGG models to demonstrate its accuracy.

## 5.2 Experiments

More precisely, experiments involve training and validating the CIFAR-10 dataset using ResNet-110 and ResNet-56 models, which have 110 and 56 layers, respectively. Similarly, the CIFAR-100 dataset is trained and validated using VGG19_bn and VGG16_bn models, which have 19 and 16 layers, respectively. Both models indisputably dominate in image classification tasks, showcasing unparalleled and highly efficient performance.

Table 1: Test accuracy (%) of proposed Caputo-based SGD on CIFAR-10 datasets for different fractional-orders $\alpha$

| Test accuracy of CIFAR - 10 | | | | | | |
|---|---|---|---|---|---|---|
| Model | | $0 < \alpha < 1$ | | | | |
| | | 0.9 | 0.8 | 0.7 | 0.6 | |
| Batch size: 128 | Resnet-110 | **94.35** | 94.27 | 93.81 | 93.75 | |
| | Resnet56 | **94.09** | 93.65 | 92.94 | 93.05 | |
| Batch size: 64 | Resnet-110 | **93.76** | 93.57 | 93.50 | 93.63 | |
| | Resnet56 | **93.98** | 93.67 | 93.57 | 93.36 | |

Table 2: Comparing test accuracies (%) of traditional optimization techniques (SGD, FCSGD_G_L, FracM) and the proposed Caputo-based SGD on CIFAR-10 datasets

| Existing methods | | | | | Proposed method |
|---|---|---|---|---|---|
| Batch size | Model | SGDM | FCSGD_G_L (Zhou et al., 2023) | FracM (Yu et al., 2022) | Caputo-based SGD |
| 128 | Resnet-110 | 93.88 | 93.15 | 94.15 | **94.35** |
| | Resnet56 | 93.83 | 92.67 | 93.57 | **94.09** |
| 64 | Resnet-110 | 93.13 | 93.69 | 93.67 | **93.76** |
| | Resnet56 | 93.34 | 93.66 | 93.35 | **93.98** |

The proposed Caputo-based SGD optimizer was employed with a fractional-order $0 < \alpha < 1$, learning rate of 0.1, and weight decay of $1e^{-5}$. The cross-entropy loss function was utilized. To train our models, we used batch sizes of 128 and 64, respectively, over a total of 200 epochs. We also used the $\phi = 0.1$ as a learning rate scheduler with a warm-up period at epochs 100 and 150. After each epoch, we evaluated the models' performance on the test set. The purpose of these experiments was to demonstrate the efficacy of the proposed Caputo-based SGD method, employing the Caputo $L_1$ scheme.

### 5.3 Results and Discussion

The results presented in our research article demonstrate the potential benefits of the method described in Algorithm 1 (please refer to Tables 1 through 4). We report our findings as a mean value based on five trials to offer a more precise estimation of the variability in classification outcomes caused by random initialization. Tables 1 and 3 provide the test accuracy for image classification tasks on the CIFAR-10 and CIFAR-100 datasets, respectively. These results are obtained by applying the proposed Caputo-based SGD for various values of the fractional-order differentiation parameter, denoted as $\alpha = 0.9, 0.8, 0.7, 0.6$, where $0 < \alpha < 1$. By adjusting the fractional order $\alpha$, the algorithm's behavior can be influenced, allowing authors to fine-tune the optimization approach to suit the specific characteristics of the problem at hand. From Table 1, we observe that the highest accuracy on the CIFAR-10 dataset is achieved when training ResNet-110/ResNet-56 models with batch sizes of 128 and 64, respectively, using $\alpha = 0.9$. In Table 3, the highest accuracy on the CIFAR-100 dataset is attained when training VGG19_bn and VGG16_bn models with a batch size of 64, using $\alpha = 0.9$.

Table 2 and Table 4 compare the accuracy of image classification tasks on the CIFAR-10 and CIFAR-100 datasets, respectively. This comparison is made between the proposed Caputo-based SGD, conventional SGD optimizers, SGD with momentum (SGDM), FCSGD_G_L (Zhou et al., 2023), and FracM (Yu et al., 2022). The results clearly indicate that the proposed Caputo-based SGD optimizer outperforms conventional optimizers and existing FGDs, achieving higher accuracy. In other words, Caputo-based SGD proves to be a more effective optimizer for image classification tasks compared to conventional SGD optimizers. This enhanced performance is likely attributable to the non-locality, degrees of freedom, and hereditary properties

Table 3: Test accuracy (%) of proposed Caputo-based SGD on CIFAR-100 datasets for different fractional-orders $\alpha$

| Test accuracy of CIFAR - 100 | | | | | |
|---|---|---|---|---|---|
| Model | | $0 < \alpha < 1$ | | | |
| | | 0.9 | 0.8 | 0.7 | 0.6 |
| Batch size: 64 | VGG19_bn | **71.15** | 70.75 | 70.32 | 69.82 |
| | VGG16_bn | **73.42** | 73.16 | 73.30 | 72.96 |

Table 4: Comparing test accuracies (%) of traditional optimization techniques (SGD, FCSGD_G_L, FracM) and the proposed Caputo-based SGD on CIFAR-100 datasets

| Existing methods | | | | | Proposed method |
|---|---|---|---|---|---|
| Batch size | Model | SGDM | FCSGD_G_L (Zhou et al., 2023) | FracM (Yu et al., 2022) | Caputo-based SGD |
| 128 | VGG19_bn | 73.00 | 71.57 | 72.16 | **73.18** |
| | VGG16_bn | 74.01 | 70.96 | 72.03 | **74.34** |

of Caputo fractional derivatives, which enable them to capture complex patterns and dependencies within the data.

## 6 Conclusion

In this study, we introduced the Caputo-based SGD algorithm for optimizing deep learning models. We incorporated a fractional order within the interval of $0 < \alpha < 1$, employing the Caputo $L_1$ numerical approximation. These techniques were used to develop the Caputo-based SGD. The proposed algorithm was then implemented in the back-propagation process to update the parameters of the input layers in DNNs. Our method capitalizes on the memory and hereditary properties of fractional calculus to address the issue of convergence to local minima and accelerate convergence towards the target value compared to traditional gradient descent methods. To evaluate the effectiveness of our approach, we conducted experiments on the CIFAR-10 and CIFAR-100 datasets using ResNet and VGG models. The results demonstrated that our method consistently achieved higher accuracy compared to existing approaches, such as SGD, FCSGD_G_L, and FracM. Additionally, our research revealed that fine-tuning the fractional order $\alpha$ can further enhance the accuracy of deep learning models. The Caputo-based SGD algorithm is particularly valuable in scenarios where long-term dependencies, noise, or anomalous behaviors significantly impact the optimization process.

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
