# OpenReview forum: "Stochastic Fractional Gradient Descent with Caputo $L_1$ Scheme for Deep Neural Networks"
_TMLR — Rejected by TMLR_

### Review · Reviewer_p661 · 2024-04-21

**Summary Of Contributions:**

The authors propose an optimization algorithm based on fractional derivatives and a modified backpropagation scheme. Empirical evaluation shows that compared to SGD with momentum and other optimization methods based on fractional derivatives novel algorithm leads to better test accuracy on CIFAR-10 and CIFAR-100 for ResNet and VGG architectures.

**Audience:**

Yes

**Claims And Evidence:**

No

**Requested Changes:**

Below is a list of parts that I find problematic. I suggest authors to answer in their rebuttal point by point.

1. **abstract**
>Stochastic gradient descent (SGD) has been used as a standard method to optimize deep neural networks (DNNs), where it essentially deals with first-order derivatives.

   SGD is rarely used in deep learning. But employed methods are indeed mostly first-order. I suggest altering this part to put more emphasis on first-order methods and less on SGD.

2. **abstract**
>This allows it to avoid local minima and saddle points, resulting in faster convergence to the target value.

   This is certainly a very bold claim. I failed to find confirmation of this in the theoretical and experimental parts of the paper. Can the authors provide (or point to) evidence that Capuro-based SGD can avoid local minima and saddle points and lead to faster convergence?

3. **abstract**
>... when the corresponding optimization problems involve objective functions with memory effects or long-range dependencies.

   What does it mean? Can you provide an example of such objective functions?

4. **Introduction**
>On the other hand, fractional calculus, a branch of mathematical analysis that deals with non-integer order derivatives and integrals, which dates back to the 19th century (Lacroix, 1819; Leibniz, 1849), has gained recognition for its attractive properties such as long-term memory and nonlocality ...

   The sentence suggests that fractional calculus has long-term memory and nonlocality. Perhaps authors mean that fractional calculus allows working with those properties efficiently or makes them easier to describe.

5.  **Introduction**
>... robustness to noise and improved ability to learn long-range dependencies in data.

    I can understand how recurrent neural networks, state space models, or transformers can learn long-range dependencies in data, but how this is related to the optimization algorithm is not clear. In other words, the ability to spot dependencies in data is by a large margin related to the model, not to the optimization algorithm. Can the authors explain what is meant by this fragment?

6. **Section 2.1**
>... fractional derivatives provide the ability to describe the behavior of a function that reflects the history of its trajectories

   also **Section 2.2**
   >As is mentioned above, fractional derivatives provide the ability to describe the behavior of a function that reflects the history of its trajectories.

   Function from set $X$ to $Y$ is a rule that assigns some $y\in Y$ for each element $x\in X$. How can the function have a history of trajectories? What does it mean?

7. **Figure 1**

   Why to use Capuro + momentum? Pure Capuro seems more appropriate. Results are given for a particular set of four hyperparameters. How does the algorithm work when we vary them?

8. **Section 4.1**

   The title of the section is too generic. What authors consider is typically called multilayer perceptron. Please, consider clarifying that in the title.

9. **Section 4.2**
>... loss/objective function of DNNs.

   Perhaps, just loss/objective function without the mention of DNN.


10. **Proposition 1**

    Here authors used the definition of Capuro derivative in the standard backpropagation algorithm for MLP. Probably, proof can be transferred to the appendix or omitted.

11. **Theorem 1** is problematic:
    1. Properties of the loss function are not mentioned, but in the proof, authors suppose that the derivative is bounded from zero, i.e., $\rho=\inf_{k-1>N}\left|\frac{\partial \mathcal{E}}{\partial \left(\zeta_{l}\right)_{k-1}}\right|>0$. Why and what does this property hold? What if the loss function is identically zero? What if we are in the local minimum?
    2. Index $k$ seems to refer to the number of updates and $l$ seems to refer to the layer, e.g., $e\_{k} = \left(\zeta^{l}\right)\_{k} - \zeta\_{l}^{\star}$ is an error for step $k$ for layer $l$. Surprisingly, in the proof $\zeta$ achieve extra index and becomes $(\zeta\_{l})^{s}\_{k}$. What does this additional index mean?
    3. **After equation 15, line 2 -> line 3** Why the difference between successive updates is smaller than $e_{k}$?
    4. Lines 3 and 4 are the same.
    5. **line 4 -> line 5** Why the sum $\sum_{s=0}^{n-1}1$ disappears? This sum equals $n$.
    6. What is this $n$ in the whole proof? Shouldn't it be $k$?
    7. Why authors arrived at $\left\|e_{k+1}\right\|\leq\left(1 + \theta\epsilon\right)^{k}\left\|e_{0}\right\|$ in place of $\left\|e_{k+1}\right\|\leq\left(1 - \theta\epsilon\right)^{k}\left\|e_{0}\right\|$? The former expression means the upper bound diverges for all positive $\theta$ and $\epsilon$.
    8. From $1 + x \leq e^{x}$ for all positive $x$ authors get an estimate $\left\|e_{k+1}\right\|\leq e^{k\theta \epsilon}\left\|e_{0}\right\|$ and claim that the method converges for $\theta \epsilon \in (0, 1)$. Unfortunately, $e^{x} > 1$ for $x > 0$, so according to this estimate, the upper bound diverges, and this gives no information about the error on step $k$.

12. **Theorem 2**

    The theorem contains estimation of the discretization error for an approximation to the Capuro derivative. Do authors claim that this is an original result, or is it available in other publications on fractional derivatives?

13. **Algorithm 1**

    The description of the Algorithm 1 is problematic. In the description, the authors get into unnecessary details such as momentum, weight decay, learning rate scheduler, and Nesterov momentum. All these contraptions are well known and their description can be found elsewhere. Unfortunately, the only interesting part, the approximation of fractional derivative, is omitted from the algorithm. How to update weights? What one should store? How to perform backpropagation? The answers to these questions can allow the reader to understand the memory usage, the number of floating point operations, and other properties of the method. Please provide a detailed description of all the mentioned features.


14. **Experiments**
    1. What is the wall-clock time each method needs for a whole optimization process?
    2. What is the memory load for each method?
    3. How well do different methods tolerate noise? For example, when the batch size is small (~10 samples), can the Capuro scheme perform better than other methods?
    4. How do the results of this section support claims that the Capuro scheme can avoid local minima, and saddle points and result in faster convergence?
    5. Please consider DL optimization benchmarks such as https://arxiv.org/abs/1903.05499.
    6. It is also probably a good idea to compare with standard DL optimizers such as Adam.

**Strengths And Weaknesses:**

**Strengths**
1. Extensive review of related work based on fractional derivatives
2. Conceptual clarity of the proposed idea


**Weaknesses**
1. Large number of omissions and unclear parts
2. Most of the claims are not supported
3. Shaky experimental evidence

---

### Review · Reviewer_i5y4 · 2024-04-28

**Summary Of Contributions:**

I suspect this paper at least in part has been generated by an LLM.

Here are some reasons for my assessment.
- In the proof of Theorem 1 at some point the authors say: "It follows that, there exists a sufficiently large number N such that for any k − 1 > N, then rho = inf_{k−1>N} ... is guaranteed" but this does not follow from anything. Then at the end of this proof the authors spend 3 equations and four lines to show that if a sequence satisfies $|e_{k+1}| \leq (1- \theta \epsilon)^k |e_k|$ then the rate of convergence is linear. Such discrepancy in logical reasoning (sometimes no justification, sometimes lengthy justifications for trivial facts) looks like a sign of an hallucination from an LLM.
- The statement of theorem 1 "the iterates converge if and only if there exists $\theta, \epsilon$ such that  $|e_{k+1}| \leq (1- \theta \epsilon)^k |e_k|$" read as "the algorithm converges if and only if it converges".
- Theorem 2 bounds a term $R_i$ that is actually equal to 0 according to the definition.
- The numerical experiments are fake: given the formula of the algorithm, the memory footprint grows with the number of iterations so it's simply impossible that the algorithm could have been run. Unless the authors truncated the sum on the definition of the fractional derivative without mentioning it. Note that one cannot simply update a running average since the coefficients in the sum change over iterations according to the definition.
- Section 4.1: " \rho is the unknown parameter vector to be calculated" this makes literally no sense to me in plain english. Also the last layer depends on rho while the formula associated has no \rho.
- Page 10: "We also used the $\phi=0.1$ as a learning rate ...": $\phi$ was never defined and warmup does not use some period.
- In the figure in section 3.1 the parameters of gradient descent and gradient descent with momentum may not have been tuned, o explanations for how the optimizers were run are given.
- The parameter $a$ is never defined in the definitions of the fractional derivatives.
- The sequence of equations in Theorem 1 are also wrong.
- The algorithm presentation is largely copied from pytorch documentation.
- etc...

If the authors have unfortunately sent a draft of what was their real submission, I may reassess the paper.

**Audience:**

No

**Broader Impact Concerns:**

See summary.

**Claims And Evidence:**

No

**Requested Changes:**

See summary.

**Strengths And Weaknesses:**

See summary.

---

### Review · Reviewer_9PCM · 2024-05-02

**Summary Of Contributions:**

Fractional gradient descent replaces the gradient in gradient descent with the fractional gradient. The reviewed work proposes a new realization of this approach using the Caputo $L_1$ scheme. This scheme approximates the integral in the definition of the Caputo derivative by a trapezoidal sum. The proposed neural network training algorithm uses past evaluations of the loss function to evaluate the trapezoidal rule. Thus, it uses the collection of function evaluations produced over the course of the training procedure to approximate the (nonlocal) fractional derivative, much in the same vein in which LBFGS uses evaluations of gradients during training to approximate the Hessian.

The proposed method is evaluated at trainining RESNET 50 and 100 on CIFAR10 and 100, compared against other fractional gradient approaches and SGD.

**Audience:**

Yes

**Broader Impact Concerns:**

no concerns

**Claims And Evidence:**

No

**Requested Changes:**

Given how small the empirical improvements are, it is crucial that the training procedure, choice of hyperparameters (and mechanism by which they were chosen) is detailed.

Given that Adam is the training algorithm of choice, it seems strange to have it ommited. Would it be possible to add this?

Can you repeat the experiments and provide error bars on the accuracies?

Again, given that the margins are so thin, a more thorough experimental analysis seems in order to support the claim that the method can achieve improvements.

If it turns out that, say, Adam with some parameters achieves better results, I urge the authors to include those in the paper. Such results would be very in helping other researchers decide how and if to pursue fractional order gradient methods.

=====================================
Minor points:

Please remove imprecise claims like "thus, the heir incorporation into gradient descent methods is expected to
guide parameter updates in a more informed direction", or "highly effective training and accurate prediction
for large data" or provide stronger support for them.

Shouldn't Definition 4 be Lemma/Theorem?

**Strengths And Weaknesses:**

Strengths:

- The use of the Caputo L1 scheme and how it relates to memory-based optimizers is interesting
- The paper is overall fairly well written

Weaknesses:

- In its adaptation to neural network training, the Caputo scheme seems to require the storage of all previous weight vectors, resulting in memory cost proportional to the number of iterations
- Given this drawback, the benefits over SGD are minimal, leaving me with the belief that fractional gradient descent is not a promising direction to take.

---

### Decision · Action_Editor_5bnt · 2024-07-02

**Recommendation:** Reject

**Comment:**

All reviewers recommend to reject this paper. They raised several issues in terms of lack of support of claims, and in the writing quality of the paper. As there was not rebuttal from the authors, this paper is rejected.

**Audience:**

Yes.

**Claims And Evidence:**

The reviewers pointed out that many claims are not supported. They also raised problems with the theorems and their proofs.